# Effect of Gossypol on Gene Expression in Swine Granulosa Cells

**DOI:** 10.3390/toxins16100436

**Published:** 2024-10-10

**Authors:** Min-Wook Hong, Hun Kim, So-Young Choi, Neelesh Sharma, Sung-Jin Lee

**Affiliations:** 1College of Animal Life Sciences, Kangwon National University, Chuncheon 24341, Republic of Korea; 2Division of Veterinary Medicine, Faculty of Veterinary Sciences & Animal Husbandry, Sher-e-Kashmir University of Agricultural Sciences & Technology of Jammu, R.S. Pura, Jammu and Kashmir 181102, India

**Keywords:** gossypol, RNA-seq, differentially expressed genes, swine, granulosa cells

## Abstract

Gossypol (GP), a polyphenolic compound in cottonseed, has notable effects on female reproduction and the respiratory system in pigs. This study aimed to discern the alterations in gene expression within swine granulosa cells (GCs) when treated with two concentrations of GP (6.25 and 12.5 µM) for 72 h, in vitro. The analysis revealed significant changes in the expression of numerous genes in the GP-treated groups. A Gene Ontology analysis highlighted that the differentially expressed genes (DEGs) primarily pertained to processes such as the mitotic cell cycle, chromosome organization, centromeric region, and protein binding. Pathway analysis using the Kyoto Encyclopedia of Genes and Genomes (KEGG) indicated distinct impacts on various pathways in response to different GP concentrations. Specifically, in the GP6.25 group, pathways related to the cycle oocyte meiosis, progesterone-mediated oocyte maturation, and p53 signaling were prominently affected. Meanwhile, in the GP12.5 group, pathways associated with PI3K-Akt signaling, focal adhesion, HIF-1 signaling, cell cycle, and ECM–receptor interaction showed significant alterations. Notably, genes linked to female reproductive function (CDK1, CCNB1, CPEB1, MMP3), cellular component organization (BIRC5, CYP1A1, TGFB3, COL1A2), and oxidation–reduction processes (PRDX6, MGST1, SOD3) exhibited differential expression in GP-treated groups. These findings offer valuable insights into the changes in GC gene expression in pigs exposed to GP.

## 1. Introduction

Cottonseed and cottonseed meal (CM) are used as protein sources for livestock. While CM is most commonly used as feed for ruminants, its use has been increasing in poultry and monogastric animal species. However, both cottonseed and CM are known to contain a toxic substance, namely, gossypol (GP), which limits the use of CM as animal feed. GP (2,2-bi(8-formyl-1,6,7-trihydroxy-5-isopropyl-3-methylnaphthalene)) is a yellowish polyphenolic compound found in the roots, leaves, stems, and seeds of the cotton plant genus Gossypium subspecies. It is known to affect male reproduction, inhibiting spermatogenesis and reducing spermatozoid motility and viability [1,2], and the use of GP as a contraceptive agent in humans has been documented [3]. GP is absorbed into the body, and it does not decompose well and continues to accumulate. Absorbed GP appears to have a long half-life and has been detected in various organs and tissues, including plasma, heart, liver, kidney, muscle, testes, and ovaries. In general, acute toxicity symptoms of GP in livestock include respiratory distress, impaired body weight gain, anorexia, weakness, apathy, and death after several days [4,5,6]. In addition, GP poisoning induces oxidative stress by promoting the formation of reactive oxygen species (ROS) and lipid peroxidation, and reduces levels of antioxidants in tissues and plasma membrane permeability [7,8]. At high concentrations, GP has also been reported to impair energy production from oxidative metabolism by interfering with the mitochondrial electron transport chain and oxidative phosphorylation [9].

Ovarian follicles are composed of oocytes surrounded by granulosa and theca cells. The growth of granulosa cells (GCs) and oocytes supports early follicle growth; in this process, GCs surround the oocytes and produce ovarian steroid hormones such as estrogen and progesterone [10]. Ovarian steroids play important roles for the normal development and function of several organs, including the brain, uterus, and mammary glands [11]. Until recently, several studies have been conducted in swine GCs. Several studies have been conducted on the production of oxidative stress (OS) and ROS by various toxic substances [12,13]. In addition, other studies have confirmed hormonal changes or cell proliferation with the addition of supplements [14,15]. Zhang et al. [16] reported that follicular atresia caused by GC apoptosis of swine may affect prepubertal reproduction. However, few studies have investigated the toxicity of GP using GCs, and among them, only changes in reproductive hormones or specific genes have been investigated [17,18,19]. In recent years, with the development of high-throughput technology capable of processing large amounts of data, and cost reduction of next-generation sequencing (NGS), sequence-based transcriptome analysis such as RNA sequencing (RNA-seq) has been used not only in humans but also in various animals and plants. RNA-seq has been shown to have considerable advantages such as examining transcriptome profiles and differentiating between different transcriptional and splicing isoforms [20,21]. Moreover, RNA-seq technology is known to be useful for analyzing expression patterns of gene populations and understanding the regulatory networks of various metabolic pathways of the analyzed genes [16].

To our knowledge, gene expression by transcriptome analysis using RNA-seq has not been confirmed in swine GCs treated with GP. Especially, previous studies investigating the effects of GP toxicity, with the goal of identifying differences in gene expression, have not been conducted in swine. Therefore, this study aims to systematically examine changes in gene expression as a result of GP treatment in swine GCs using RNA-seq.

## 2. Results

### 2.1. GP Exposure Affected the Number of Viable Swine GCs

To evaluate changes in cellular viability as a result of GP treatment, cell viability assays were performed after GP treatment at different concentrations for 72 h using cell proliferation assays (Figure 1a). Our results showed that the viability of the GCs significantly decreased (*p* < 0.001) with GP treatment (6.25 to 25 µM) compared with the control (Ctrl). GCs cultured with 12.5 µM GP (GP12.5) showed cellular viability levels that were close to the IC50 concentration (i.e., the half maximal inhibitory concentration) at 51.2%. Based on the above results, we observed the cell morphology of the Ctrl, GP6.25 (GCs treated with 6.25 µM GP) and GP12.5 using a microscope (Figure 1b–d). As the concentration of GP increased in vitro for 72 h, the number of dead and floating cells increased and the status of adherent cells deteriorated compared to the Ctrl group. Taken together, these results suggest GP has a detrimental effect on GCs.

### 2.2. Overview of Sequencing Data Using RNA-Seq Analysis

All three groups, GP6.25, GP12.5, and Ctrl, consisted of two cDNA libraries each. The mapped reads and annotation showed that the three libraries were of high quality, with a total of 20,526,910, 17,897,687, and 20,352,139 sequences that were matched, respectively (Table 1). The reads were obtained for about 88.33% of the swine reference genome (*Sus scrofa*) in NCBI. In addition, the Q30 range was determined to be from 89.5% to 92.1%. These findings suggest that the library quality for each group was good and suitable for analysis. The results of the hierarchical clustering of DEGs between the three groups showed a clear discrimination, which is presented as a heat map in Figure 2a. The distribution of FPKM for all the samples was indicated as a similar expression level, and most of the mRNAs detected in the RNA-seq analysis were also found to be protein-coding genes (Figure 2b).

### 2.3. Differential Gene Expression of GCs by GP Exposure Using RNA-Seq Analysis

The numbers of genes found in each library group were 14,344, 14,294, and 14,482, respectively (Table 1). Among them, a total of 390 and 951 significantly different genes were obtained from the GP6.25 and GP12.5 groups with the conditions of *p* < 0.05, FDR < 0.05, and |FC| ≥ 2 (Figure 2c). In the GP6.25 group, compared with the Ctrl group there were 94 upregulated DEGs and 296 downregulated DEGs, while in the GP12.5 group, there were 342 upregulated DEGs and 609 downregulated DEGs. A total of 294 genes were identified, including 58 upregulated and 236 downregulated genes in intersection portion of both concentrations (IPC). Table 2 shows the most upregulated and downregulated genes overlapping with the GP6.25 and GP12.5 groups.

Functional classification of the top 30 GO terms for all DEGs of the three groups (GP6.25, GP12.5, IPC) is shown in Figure 3. In the biological process category, the GO terms found to be enriched in the DEGs were those related to ‘mitotic cell cycle’ and ‘mitotic cell cycle process’ for all groups. In the cellular component category, the enriched GO terms of interest included ‘chromosome, centromeric region’ and ‘kinetochore’ in the GP6.25 group and IPC group. However, the GO terms found to be the most enriched in the GP12.5 group were ‘extracellular region part’, ‘extracellular region’, and ‘extracellular matrix’. In the molecular function category, the GO terms found to be enriched in the DEGs were centered on ‘binding’ and ‘protein binding’ for all groups. Each of the 170, 423, and 127 DEGs were then assigned to KEGG annotations in the GP6.25, GP12.5, and IPC groups. Our results indicated that 8, 30, and 8 enriched pathways were significantly changed (*p* < 0.05) in each group, respectively (Figure 4). The GP6.25 and IPC groups revealed that eight pathways including cell cycle, oocyte meiosis, progesterone-mediated oocyte maturation, and p53 signaling pathways, were the most changed. In the GP12.5 group, it was revealed that 30 pathways including the PI3K-Akt signaling pathway, focal adhesion, HIF-1 signaling pathway, cell cycle, and ECM–receptor interaction were the most changed.

### 2.4. Validation of Selected Genes by qRT-PCR

Based on sequencing data and functional analysis, 11 genes of interest, including those associated with female reproductive functions (*CDK1*, *CCNB1*, *CPEB1*, and *MMP3*), cellular component organization (*BIRC5*, *CYP1A1*, *TGFβ3*, and *COL1A2*), and oxidation–reduction processes (*PRDX6*, *MGST1* and *SOD3*), were selected for validation by qRT-PCR. As shown in Figure 5, five genes (*CYP1A1*, *CPEB1*, *MMP3*, *PRDX6*, and *MGST1*) were found to be significantly upregulated, and six genes (*CDK1*, *CCNB1*, *BIRC5*, *TGFβ3*, *COL1A2*, and *SOD3*) were significantly downregulated in both groups treated with GP (GP6.25, GP12.5). These results were consistent with those obtained from RNA-seq. However, three genes (*CYP1A1*, *PRDX6*, and *SOD3*) were not found to be significantly changed in the GP6.25 group. Overall, our results indicated that RNA-seq generated a reliable dataset for the transcriptome analysis of GP-treated GCs.

### 2.5. GP Cytotoxicity Induced Various Changes in GCs’ Protein Expression

Based on the mRNA expression results of qRT-PCR and RNA-seq, we selected four proteins for further western blot analysis. The expression of proteins related to female reproductive function (CCNB1), cellular component organization (MMP3), and oxidation–reduction processes (MGST1, PRDX6) are shown in Figure 6. In our results, the expression levels of three proteins significantly changed (increased or decreased) according to the difference in GP concentration, with the exception being the MGST1 protein. The expression of the MGST1 protein was not significant, but an upregulated pattern was identified. Uncropped western blot images of Figure 6 are shown in Appendix A.

## 3. Discussion

In recent years, RNA-seq, as a method for the sequence-based analysis of transcriptomes, has become a useful tool for analyzing gene expression patterns with high-throughput NGS technology. In addition, RNA-seq allows for the analysis of complex whole eukaryotic transcriptomes with higher reproducibility, wider dynamic range, less bias, and lower frequency of false positive signals compared with traditional cDNA microarray technologies [22]. An earlier study has reported that the Pearson correlation between RNA-seq and qRT-PCR can reach > 0.9 [23], suggesting that RNA-seq is an advanced technology that is reliable and reproducible. In addition, several studies using RNA-seq have been conducted on the swine transcriptome such as the effects of ageing or various antioxidants in GCs [24,25,26]. However, to date, there exist limited studies that have investigated the toxic effects of GP in swine, and the majority of these studies have focused on differences in internal organ size and changes in reproductive hormones [4]. As such, the effects of GP toxicity on genes in GCs are currently almost unknown. Therefore, in this study, we investigated the transcriptome profile of the swine GCs using RNA-seq, with the aim of identifying candidate genes for major functional changes caused by GP cytotoxicity.

Previous studies have shown that cellular proliferation is affected by GP toxicity. Indeed, GP has been shown to inhibit cell growth in a dose-dependent manner in various species, including human. The majority of the studies evaluating the toxic effects of GP at the cellular level have been performed on cancer cells, including those of leukemia [27], prostate cancer [28], breast cancer [29], ovarian cancer [30], carcinoma [31], and other malignancies [32], with the focus on preventing cancer metastasis. As a result, it has been reported that the proliferation of cancer cells is inhibited by GP toxicity. However, some studies using primary cells have shown a pattern of cell growth caused by GP toxicity different to that in cancers. Lin et al. [19] reported that GP at 10, 50, and 150 µg/mL significantly decreased the number of bovine oocytes undergoing cumulus expansion cultured at 24 h, by 15%, 50%, and 80%, respectively. In another study, however, cellular proliferation was shown to be significantly increased by GP at 5 and 25 µg/mL in swine GCs cultured for 44 h [33]. In this study, we found that the number of viable cells was dramatically decreased in a dose-dependent manner in GP-treated swine GCs. A possible reason for the discrepancy between the results of the present study and those from previous studies using primary cells may be related to differences in the culture period. However, Hsiao et al. [34] reported that the population of apoptotic cells increased significantly with increasing GP dose, and the cells were more likely to enter late apoptosis and undergo cell arrest compared with the control group. Collectively, our results suggested that the inhibition of proliferation by GP toxicity may occur after exposure to GP for a sufficient period of time.

In females, granulosa and theca cells and oocytes are the constituents of mammalian ovarian follicles. GCs play a complex and central role in the development of the ovarian follicle. During this process, GCs surround and nurse the oocyte, supporting its maturation. In this study, we identified DEGs from an RNA-seq library of swine GCs treated with two GP concentrations (6.25 and 12.5 µM). These DEGs were then divided into three groups (GP6.25, GP12.5, and IPC), and we investigated the differences in gene expression caused by GP toxicity through GO and KEGG pathway analyses. We found that the identified DEGs are associated with processes related to cellular component organization, oxidation–reduction, and female reproductive functions. From these results, we selected 11 genes to investigate differences in mRNA and protein expression as a result of GP exposure.

Meiotic progression and oocyte maturation are regulated by activity of the universal cell cycle regulator called the maturation-promoting factor (MPF). In addition, MPF is important for the precise regulation of mitosis and enables chromatin condensation and alignment through histone phosphorylation regulation. As such, MPF, which plays an important role in the cell cycle, is formed primarily through the interaction of the regulatory subunit, cyclin B1 (CCNB1), with the catalytic subunit, cyclin-dependent kinase 1 (CDK1) [35]. Adhikari et al. [36] reported that the *CDK1* gene is necessary to promote the resumption of meiosis. Jiang et al. [37] reported that cyclin and CDK accelerate cell cycle progression, whereas CDK inhibitors slow the progression. CCNB1 is known as a major protein involved in the regulation of oocyte maturation in mammals. The synthesis of CCNB1 is necessary for germinal vesicle breakdown (GVBD) induction in the second meiosis [38,39]. Stanley et al. [40] reported that the levels of cyclin A, B1, and CDK1 were reduced by hexavalent chromium in rat GCs, delaying cell entry and progression through the G2-M phase. In the present study, the mRNA expression of *CDK1* and *CCNB1* was decreased in a dose-dependent manner in GP-treated GCs, as observed from both RNA-seq and qRT-PCR. We found that protein expression of CCNB1 also decreased dose-dependently. These results suggest that GP indirectly inhibits the maturation and development of oocytes by delaying cell cycle progression by reducing the levels of *CDK1* and *CCNB1* in swine GCs.

Translational activation and polyadenylation of cell cycle regulators including cyclins and CDK are found primarily in the 3′-UTR of the mRNAs; the cytoplasmic polyadenylation element (CPE) is located in these sequences [41]. The CPE binds to CPE-binding protein 1 (CPEB1), CPEB1 is involved in translational activity or inhibition, according to the phosphorylation state [42]. In oocytes, the non-phosphorylated CPEB1 protein is known to interact with the Maskin and Pumilio proteins, which are involved in translational repression [43,44]. In previous studies of various species, including human, CPEB has been reported to be involved in both the inhibition and stimulation of *CCNB1* mRNA. In this regard, Nakahata et al. [43] reported that the overexpression of Pumilio in *Xenopus* oocytes leads to inhibition of *CCNB1* levels. Taken together, the phosphorylation of CPEB1 suggests that it may be involved in the regulation of CCNB1 activity. In our results, the mRNA expression of *CPEB1* was upregulated and that of *CCNB1* was downregulated. We have not been able to identify exactly how GP acts on the phosphorylation of CPEB1. However, our results suggested that Pumilio bound to the *CPEB1* gene likely regulated the levels of CCNB1. As a result, GP is involved in the expression of *CPEB1* and *CCNB1* genes in GCs; thus, it seems to affect the growth and maturation of oocytes.

In general, the extracellular matrix (ECM) is a natural substrate that supports the function of cells such as intercellular communication, adhesion, migration, and proliferation. In particular, it affects cell morphology, follicle development, aggregation, communication, and steroidogenesis in the ovary. ECM metabolism, which causes periodic changes throughout the reproductive cycle, is caused by the action of certain types of proteins known as matrix metalloproteinases (MMPs). MMPs are enzymes that degrade ECM proteins throughout the body to facilitate tissue remodeling [45]. Many studies have shown that MMPs play an important role in follicle development, ovulation, luteinization, and luteolysis, associated with follicular ECM alterations [46]. Zhu et al. [47] reported that the gene expression of *MMP1*, *MMP3*, and *MMP9* was significantly increased in the ovary during sexual maturation in chicken. In addition, Hui et al. [24] reported that *MMP3* expression in GCs decreased significantly during ageing in swine. In this study, we identified that the expression of the MMP3 protein and mRNA were significantly increased in GP-treated GCs in a dose-dependent manner. Kuittinen et al. [48] reported that high *MMP* expression was found in lymphoid and malignant transformed cells. In summary, our results suggested that the upregulation of *MMP3* in GCs is the result of GP cytotoxicity rather than the maturation of the cells.

Survivin (BIRC5) is an inhibitor of apoptosis and has been shown to play an important role in cellular apoptosis by blocking the downstream step of mitochondrial cytochrome c release [49]. Several studies have shown that the BIRC5 protein is an inhibitor of apoptosis and is overexpressed in human cancer cells [50,51]. Jiang et al. [37] reported that the toxicity of GP downregulated the mRNA expression levels of, e.g., *BIRC5*, involved in the cell cycle and survival, thereby inhibiting the proliferation of prostate cancer cells. Our mRNA expression results showed that BIRC5 was downregulated dose-dependently as a result of GP cytotoxicity. And these results indicated that GP induces apoptosis by inhibiting cell proliferation of swine GCs.

The catalytic reaction of cytochrome P450 proteins (CYPs) is not essential for the maintenance of cellular activity, but plays an important role in the metabolism and removal of exogenous substances [52]. Enzymes belonging to the CYP family are called phase I enzymes; they monooxygenase, reduce, and hydrolyze various substances such as lipids, steroidal hormones, and xenobiotics [53]. Phase I enzymes are expressed primarily in the liver, but they are also present in other tissues such as uterus, placenta, brain, kidney, and testis [54]. Previous studies have reported that certain CYP isoforms (CYP1A1, CYP1A2, and CYP2B) are present in swine prepubertal ovary cells [55,56]. The Ah receptor (AhR) is a transcription factor that mediates the toxic effects of chemicals. Pocar et al. [57] reported that treatment with TCDD (2,3,7,8 tetrachlorodibenzo-p-dioxin) and PCB126 (polychlorinated biphenyl 126) induced the AhR signal transduction response via the expression of *CYP1A1*, an AhR target gene in swine thyrocyte. In addition, Barć et al. [52] reported that the activity and expression of *CYP1A1* dose-dependently increased in polychlorinated naphthalene-treated swine GCs. According to the qRT-PCR results in the present study, *CYP1A1* mRNA was significantly increased in GP-treated GCs, which is likely due to GP cytotoxicity. However, the RNA-seq results did not show a significant difference in *CYP1A1* mRNA expression with GP treatment, although an upregulation pattern was observed.

The transforming growth factor β (*TGFβ*) gene is involved in several important cellular functions, including proliferation and differentiation. It is also a molecular marker related to the embryonic growth and development of mammalian oocytes [58]. Dragovic et al. [59] reported that the *TGFβ* gene is an important marker of cumulus expansion, and this process has been shown to be determined by the TGFβ protein signaling pathway. *COL1A*2 is involved in TGFβ signaling, which plays an important role in mediating ovarian age and oocyte quality [60,61]. The *COL1A2* gene is involved in cellular proliferation and regulation of the cell cycle, and decreased levels of *COL1A2* in old mares indicate the potential deregulation of TGFβ signaling as an underlying factor in the age-related decline in oocyte quality [62]. In the present study, we found that the mRNA expression of the *COL1A2* and *TGFβ3* genes was significantly downregulated. In view of the function of these two genes, these results suggest that GP inhibits the proliferation and differentiation of swine GCs, and thus, could affect the growth and development of oocytes.

To evaluate the effect of GP toxicity on intracellular oxidative damage in swine GCs, the relative mRNA and protein abundance of three genes related to ROS production was analyzed. Peroxiredoxins (PRDXs) are peroxidases that work together to detoxify ROS and provide cytoprotection from internal and external environmental stress [63,64]. Kubo et al. [65] showed that *PRDXs* are downregulators of various apoptotic pathways. In particular, since cumulus–oocyte complexes form a functional entity, the upregulation of PRDX6 in oocytes at the protein level and the accumulation of deadenylated transcripts have been shown to play an important role not only in embryo development but also oocyte maturation [66]. Microsomal glutathione transferase 1 (MGST1) is a member of the membrane-associated proteins in the eicosanoid and glutathione metabolism (MAPEG) superfamily [67]. MGST1 is a membrane protein located in the endoplasmic reticulum and the outer mitochondrial membrane that protects cells from oxidative stress [68]. Several studies have reported that MGST1 is activated by a variety of factors, including reactive oxygen (H_2_O_2_) and nitrogen (ONOO¯, NO) species, sulfhydryl reagents, proteolysis, heating, radiation, and dimerization of the homotrimers [69,70]. In addition, MGST1 has been shown to play an important role in the detoxication of drugs and xenobiotics and may protect membranes against lipid peroxidation [71,72]. SOD3 is an important antioxidant enzyme located in the ECM of numerous tissues and the glycocalyx of cell surfaces. SOD3 removes superoxide and ROS to prevent cell death and protect normal tissues [73]. In addition, SOD3 may play a critical role in the protection against external environmental factors such as microbiological pathogens [74]. Basina et al. [33] reported that GP stimulates O_2_ generation by SOD activity inhibition in swine GCs. The authors proposed that the increased O_2_ levels play a role in reducing steroid production in GCs. In this study, we found that the *SOD3* gene was downregulated according to the GP concentration. This result is consistent with the SOD activity reported by Basini et al. [33]. Furthermore, in our study, the *MGST1* and *PRDX6* genes were found to be upregulated at both the mRNA and protein levels. Collectively, these results indicate that these genes may be part of the cellular defense against oxidative stress caused by GP toxicity in the differentiation process of GCs.

In conclusion, we used RNA-seq to examine the gene expression profiles of GP-treated swine GCs. To our knowledge, this is the first time that the effects of GP cytotoxicity in swine GCs have been studied using RNA-seq. We selected a total of 11 genes and classified their functions using GO and KEGG analyses of the DEGs obtained with GP treatment. qRT-PCR and western blot analysis showed significant differences in the expression of selected genes. However, further studies are warranted to identify the mechanisms underlying the functions of each gene. The results of this study provide a basis for examining the effects of GP toxicity in swine. In addition, our findings provide greater insight into the various aspects of GP cytotoxicity in GCs.

## 4. Materials and Methods

### 4.1. Isolation and GCs Primary Culture

Prepubertal gilt ovaries were obtained from a local slaughterhouse and transported to the laboratory within 2 h of isolation in a vacuum thermos flask in 0.9% sterile physiological saline at 37 °C. Follicular fluid and GCs were aspirated from follicles with diameters of 3–6 mm that contained clear follicular fluid using a 10 mL syringe with an 18-gauge needle. The GCs were then transferred to a 15 mL centrifuge tube and centrifuged at 500× *g* for 5 min for precipitation. The precipitated cells were resuspended with ammonium chloride (NH_4_Cl) at 37 °C for 1 min to remove red blood cells. The cells were then washed twice with phosphate-buffered saline (PBS) containing 1% penicillin–streptomycin solution. GCs were cultured in DMEM/F12 (Gibco, Carlsbad, CA, USA) medium supplemented with 10% FBS (Gibco) and 1% penicillin–streptomycin solution in a humidified incubator with 5% CO_2_ at 37 °C for 3 days. When a complete monolayer had formed in the primary culture, the GCs were washed with PBS, trypsinized, and harvested for additional experiments.

### 4.2. Subculture and GP Treatments

Stock solutions of GP were prepared by first dissolving GP in DMSO at a 10 mM concentration and they were stored in aliquots at −20 °C until further use. The stock solution was then diluted to concentrations of 0.625, 1.25, and 2.5 mM in DMSO before the start of the experiments. For the experiments, the diluted GP was further diluted (1 in 100) when added to the cell culture medium. The cells obtained from the primary culture were subcultured into wells or dishes in a suitable amount for each experiment, with 1 × 10^6^ cells/dish in 100 mm culture dishes (SPL Life science, Pocheon, Korea) for RNA-seq, 1 × 10^5^ cells/well in 6-well culture plates (SPL Life science) for qRT-PCR and western blot analysis, and 0.5 × 10^4^ cells/well in 96-well culture plates (SPL Life science) for cellular proliferation assays. All culture plates and dishes were incubated at 5% CO_2_ at 37 °C. After seeding the cells for 24 h, to study the cytotoxic effects of GP in GCs, GP was added to the medium at final concentrations of 6.25, 12.5, and 25 µM and the cells were incubated for 72 h. The GP-untreated group (Ctrl) was incubated with only DMSO at the same concentrations as the GP treatment groups to ensure the accuracy of the experimental design. The experiment was repeated three times with identical conditions to provide three biological replicates for each time point and treatment.

### 4.3. Measurement of Cell Proliferation

GCs (0.5 × 10^4^ cells/well) were seeded with 200 µL of culture medium in 96-well plates. After a culture period of 24 h, the cells were treated with various final concentrations of GP (6.25 to 25 µM) for 72 h, while DMSO-treated cells without GP treatment served as the Ctrl group. The proliferation of GCs was measured at 570 nm using a microplate reader with the CellTiter 96 Non-Radioactive Cell Proliferation Assay (Promega, Madison, WI, USA) according to the manufacturer’s instructions.

### 4.4. RNA Extraction, Library Construction, and Sequencing Analysis

After GP treatment, total RNA was extracted from the GCs using Trizol reagents (Invitrogen, Carlsbad, CA, USA). RNA extraction was performed in duplicate for each sample. The obtained total RNA was assessed using the Agilent 2100 Bioanalyzer (Agilent technologies, Santa Clara, CA, USA). The Illumina TruSeq Stranded mRNA Sample Preparation kit (Illumina, San Diego, CA, USA) was used for preparing an mRNA sequencing library from the isolated total RNA. All libraries were quantified by qPCR using a CFX96 real-time PCR detection system (Bio-Rad, Hercules, CA, USA) and sequenced with a paired-end 75 bp plus single 8 bp index reads using a NextSeq 500 sequencer (Illumina). Potentially existing sequencing adapters and low-quality bases (5′ and 3′ ends) in the raw reads were trimmed by the Skewer software [75]. The cleaned high-quality reads, after trimming the low-quality bases and sequencing adapters, were mapped to the reference genome by the STAR software [76]. The calculation of Q30, GC content, and sequence duplication levels were based on the clean reads using FastQC (http://www.bioinformatics.babraham.ac.uk/projects/fastqc/).

### 4.5. Identification of DEGs

The mapped reads for gene expression values were assembled with Cufflinks following the swine reference genome (*susScr11*) annotation [77]. The gene annotation of the reference genome from the UCSC genome (https://genome.ucsc.edu) in GTF format was used as a gene model and the expression values were calculated in the fragments per kilobase of transcripts per million mapped reads (FPKM) unit. The differentially expressed genes (DEGs) between the two selected biological conditions were analyzed by the Codify software in the Cufflinks package (version 2.2.1) [78]. To compare the expression profiles among the samples, the normalized expression values of the selected few hundred DEGs were unsupervised-clustered by in-house R scripts.

### 4.6. GO Enrichment and KEGG Pathway Enrichment Analysis

To gain insight into the biological and functional roles of the DEGs, gene set overlapping between the analyzed DEGs and functional categorized genes, including biological processes of Gene Ontology (GO), Kyoto Encyclopedia of Genes and Genomes (KEGG, www.kegg.jp/kegg/kegg1.html) pathways, and other functional gene sets by g:Profiler (https://biit.cs.ut.ee/gprofiler), was performed [79]. A corrected *p*-value of 0.05 was set as the threshold to identify significantly different pathways [80] and *p*-values were adjusted using the Benjamini–Hochberg method [81].

### 4.7. Quantitative RT-PCR

cDNA was reverse transcribed from 500 ng of total RNA using an iScript cDNA synthesis kit (Bio-Rad), while quantitative RT-PCR (qRT-PCR) analyses were performed using an ABI 7500 real-time PCR system (Applied Biosystems, CA, USA). PCR was conducted in a 20 µL reaction volume with PowerUP SYBR Green Master Mix (Thermo Fisher Scientific, Waltham, MA, USA) and 0.4 pmol of specific primer pairs (Appendix A). The general qRT-PCR reaction conditions were as follows (except *MMP3* and *SOD3* genes): 2 min at 50 °C, 2 min at 95 °C, followed by 45 cycles of 15 s at 95 °C and 1 min at 60 °C. The qRT-PCR conditions of the *MMP3* gene were as follows: 2 min at 50 °C, 10 min at 95 °C, followed by 45 cycles of 30 s at 95 °C, 30 s at 59 °C, and 30 s at 72 °C, with a final extension of 10 min at 72 °C. The qRT-PCR conditions for the *SOD3* gene were as follows: 2 min at 50 °C, 10 min at 95 °C, followed by 50 cycles of 30 s at 95 °C, 30 s at 59 °C, and 30 s at 72 °C, with a final extension of 10 min at 72 °C. The standard curve of the *β-actin* gene in swine GCs was used as the reference to normalize the mRNA expression of the gene of interest. The relative expression of the genes was calculated using the 2^−ΔΔCt^ method. Three separate experiments were performed on different cultures and each sample was assayed in triplicate.

### 4.8. Western Blot Analysis

Protein lysates isolated from GCs were used for western blot using the PRO-PREP Protein Extraction Solution (Intron Biotechnology, Seongnam, Korea) according to the manufacturer’s procedure. Protein concentrations were determined using a Pierce BCA Protein Assay Kit (Thermo Fisher Scientific). Samples containing 10 µg of protein were boiled in Laemmli buffer and resolved on 12% Tris-glycine polyacrylamide gels, and then, transferred to polyvinylidene fluoride (PVDF) membranes (Millipore Corporation, Bedford, MA, USA). The membranes were blocked in 5% skim milk and incubated with primary antibodies as follows: anti-MGST1 (MyBioSource, San Diego, CA, USA), 1:500; anti-PRDX6 (MyBioSource), 1:1000; CCNB1 (Aviva Systems Biology, San Diego, CA, USA), 1:1000; MMP3 (Biorbyt, Cambridge, UK), 1:250; β-actin (Biorbyt), 1:500 for 2 h at RT. The membranes were washed in TBST, and then, incubated with secondary antibody (mouse anti-rabbit IgG-HRP, Santa Cruz, CA, USA; 1:2500) for 1 h at RT. Immunoreactive bands were detected using the Pierce ECL Plus Western Blotting Substrate (Thermo Fisher Scientific) in conjunction with the ImageQuant LAS 500 imager system (GE Healthcare, Little Chalfont, UK). Relative sample intensities were computed by scanning and quantifying the immunoblot data using the ImageJ software (National Institutes of Health, Bethesda, MD, USA) [82]. Each western blot experiment was performed three times and representative image results are shown. Semiquantitative results are reported as the mean ± SEM.

### 4.9. Statistical Analysis

Differences between the Ctrl and GP-treated groups were statistically determined using one-way analysis of variance. Results are presented as mean ± SEM of at least three separate experiments performed on different cultures. All statistical analyses were performed using the SAS software ver. 9.4 (SAS institute Inc., NC, USA). Results were considered statistically significant at *p* < 0.05. Frequency distribution graphs were plotted using GraphPad Prism ver. 7.0 (GraphPad software Inc., San Diego, CA, USA).

## Figures and Tables

**Figure 1 toxins-16-00436-f001:**
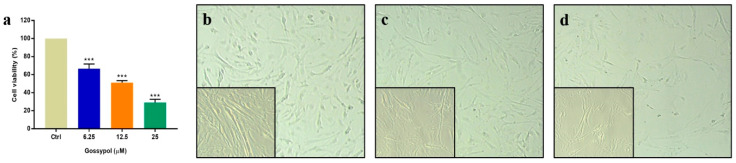
Effect of GP on the viability of swine GCs *in vitro*. Swine GCs were cultured with 0, 6.25, 12.5, and 25 µM GP for 72 h. (**a**) GC viability at different concentrations (6.25–25 µM) of GP. Data are expressed as mean ± SEM (n = 3) *** *p* < 0.001. (**b**–**d**) present the morphology of cultured GCs treated with GP. Large pictures were captured at 40× magnification, while small pictures were captured at 100× magnification. (**b**) GCs cultured in 0 µM GP (non-treated). (**c**) GCs cultured in 6.25 µM GP. (**d**) GCs cultured in 12.5 µM GP. GP: gossypol; GCs: granulosa cells; Ctrl: GP-untreated GCs.

**Figure 2 toxins-16-00436-f002:**
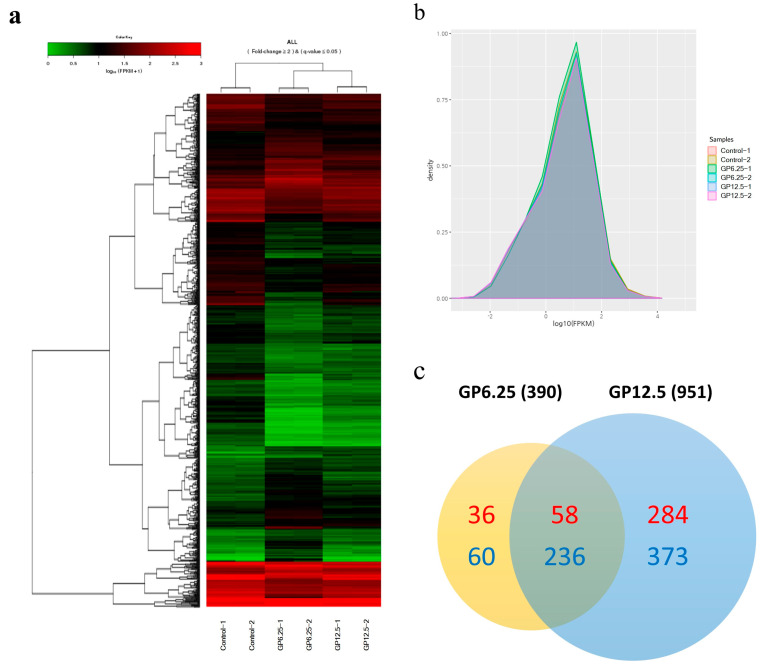
Summary of RNA−seq mapping data. (**a**) Comparative expression of selected transcriptomes across the GP−treated and Ctrl groups. Heatmap for the expression values in log10 (FPKM) units of the selected DEGs. Red indicates upregulated, while green indicates downregulated gene expression (*p* < 0.05). (**b**) Density plot for log10 (FPKM) values to compare gene expression between each sample. (**c**) Venn diagram showing the number of differently expressed genes between the GP6.25 and GP12.5 groups (*p* < 0.05). Red color: upregulated, blue color: downregulated; Ctrl: GP-untreated GCs, GP6.25: GCs treated with 6.25 µM GP; GP12.5: GCs treated with 12.5 µM GP; GP: gossypol; GCs: granulosa cells.

**Figure 3 toxins-16-00436-f003:**
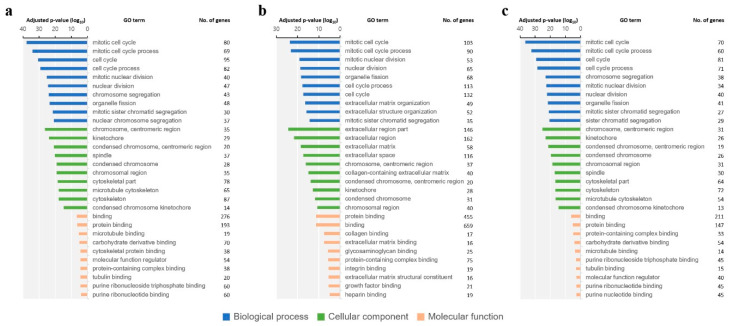
GO functional analysis of DEGs. GO term enrichment analysis results based on the different GP treatments were retrieved and compared with the Ctrl group using g:Profiler (https://biit.cs.ut.ee/gprofiler). The 10 most significantly (*p* < 0.05) enriched GO terms related to biological processes, cellular components, and molecular functions are shown. All adjusted statistically significant values of the terms were −log10 converted. (**a**) GO term enrichment result of the DEGs in the GP6.25 group. (**b**) GO term enrichment result of the DEGs in the GP12.5 group. (**c**) GO term enrichment result of the DEGs in the IPC group. GO: Gene Ontology; GP: gossypol; Ctrl: GP-untreated GCs; GP6.25: GCs treated with 6.25 µM GP; GP12.5: GCs treated with 12.5 µM GP; IPC: intersection portion of both concentrations; DEGs: differentially expressed genes.

**Figure 4 toxins-16-00436-f004:**
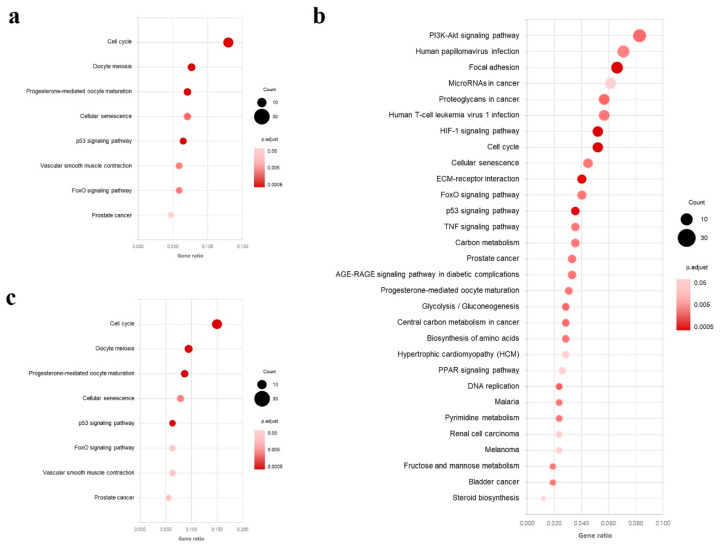
A scatter plot of the significantly enriched KEGG pathways (*p* < 0.05, www.kegg.jp/kegg/kegg1.html) of the DEGs based on transcriptome sequencing analysis of each GP-treated group compared with the Ctrl group. The y-axis represents the name of the pathway, while the x-axis represents the gene ratio. Dot size represents the number of genes and the color indicates the *p*-value. (**a**) Enriched KEGG pathways of the DEGs in the GP6.25 group. (**b**) Enriched KEGG pathways of the DEGs in the GP12.5 group. (**c**) Enriched KEGG pathways of the DEGs in the IPC group. KEGG: Kyoto Encyclopedia of Genes and Genomes; DEGs: differentially expressed genes; GP6.25: GCs treated with 6.25 µM GP; GP12.5: GCs treated with 12.5 µM GP; IPC: intersection portion of both concentrations.

**Figure 5 toxins-16-00436-f005:**
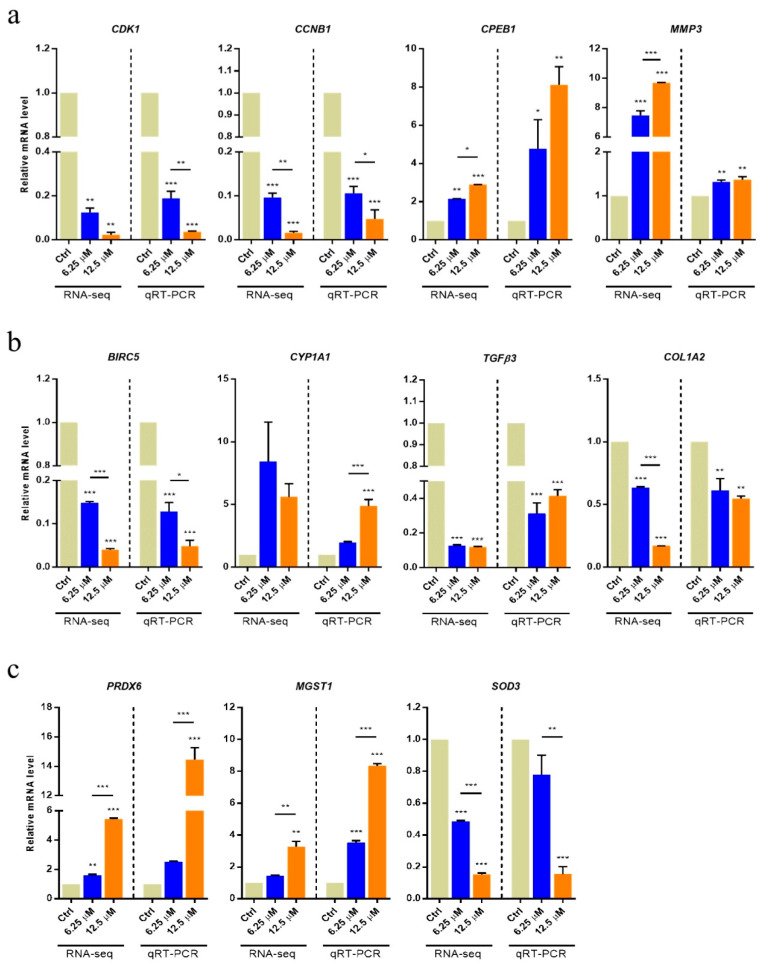
Quantification of the mRNA profile of GP-treated GCs (6.25 and 12.5 µM) using RNA-seq and qRT-PCR. GCs incubated with each GP concentration for 72 h. GP exposure affected the mRNA abundance of certain genes in the cells. (**a**) The cellular component organization of related genes (*BIRC5*, *CYP1A1*, *COL1A2*, and *TGFβ3*) was measured and presented. (**b**) The female reproductive function-related genes (*CDK1*, *CCNB1*, *CPEB1*, and *MMP3*) measured and presented. (**c**) Oxidation–reduction process-related genes (*PRDX6*, *MGST1*, and *SOD3*) were measured and presented. The mRNA levels of all genes were normalized to the swine *β-actin* gene. Results are presented as mean ± SEM. All experiments were repeated at least three times. Ctrl: GP-untreated GCs; GP: gossypol; GCs: granulosa cells; * *p* < 0.05, ** *p* < 0.01, *** *p* < 0.001.

**Figure 6 toxins-16-00436-f006:**
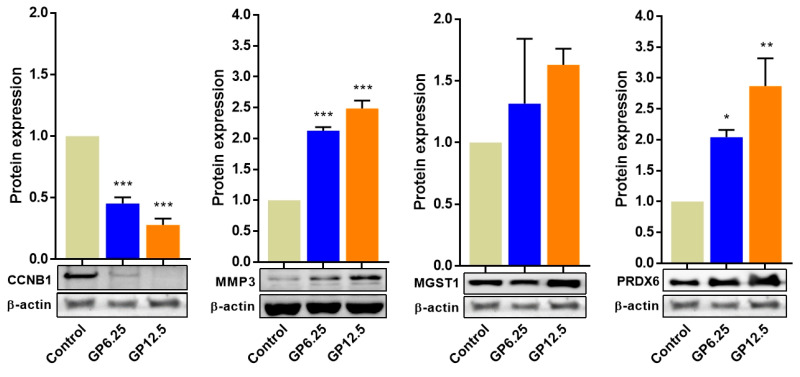
Western blot analysis of CCNB1, MMP3, MGST1, and PRDX6 proteins with different concentrations of GP (6.25 and 12.5 µM) in swine GCs. The protein levels were normalized to β-actin. Data are expressed as the mean ± SEM. All experiments were repeated at least three times. Full-length images of the above cropped images are presented in Appendix A. Ctrl: GP-untreated GCs; GP: gossypol; GCs: granulosa cells; *: *p* < 0.05, **: *p* < 0.01, ***: *p* < 0.001.

**Table 1 toxins-16-00436-t001:** Summary statistics of RNA-sequencing data obtained from GP-treated (6.25 and 12.5 µM) and untreated swine GCs.

	Ctrl	GP6.25	GP12.5
Total reads	20,529,934	17,898,643	20,353,291
Mapped reads	20,526,910	17,897,687	20,352,139
Yield (Mb)	3119	2720	3092
Q30	91.9	89.5	92.1
Mapping rate (%)	89.83	84.50	90.65
Unique mapped (%)	17,843,415 (86.93)	14,844,050 (82.94)	17,995,481 (88.42)
Multiple mapped (%)	597,953 (2.91)	405,462 (2.27)	450,752 (2.21)
Number of detected genes	14,344	14,294	14,482

GCs: granulosa cells; Ctrl: GP-untreated GCs; GP6.25: GCs treated with 6.25 µM GP; GP12.5: GCs treated with 12.5 µM GP.

**Table 2 toxins-16-00436-t002:** Top 10 upregulated and downregulated genes in swine GCs treated with different concentrations (6.25 and 12.5 µM) of GP compared with Ctrl group.

Genes	Description	Log2 Fold Change		*p*-Value	
		GP6.25	GP12.5	GP6.25	GP12.5
UP					
*SLC2A5*	Solute carrier family 2 member 5	1.232	4.250	0.00005	0.00005
*SLC16A3*	Solute carrier family 16 member 3	1.385	3.279	0.0004	0.00005
*MMP3*	Interstitial collagenase 18 kDa interstitial collagenase	2.896	3.279	0.00005	0.00005
*SCD*	Acyl-CoA desaturase	1.113	3.205	0.00005	0.00005
*PPP1R3C*	Protein phosphatase 1 regulatory subunit 3C	2.129	2.869	0.00005	0.00005
*IL1RL1*	Interleukin 1 receptor-like 1	1.276	2.815	0.00005	0.00005
*GPNMB*	Transmembrane glycoprotein NMB precursor	1.710	2.740	0.00005	0.00005
*SNTB1*	Syntrophin beta 1	1.356	2.566	0.00005	0.00005
ENSSSCG00000027013	-	1.565	2.400	0.00005	0.00005
*TFRC*	Transferrin receptor protein 1	1.724	2.314	0.00005	0.00005
DOWN					
*TOP2A*	DNA topoisomerase 2-alpha	−2.442	−8.242	0.00005	0.00255
ENSSSCG00000002849	-	−2.731	−7.591	0.00005	0.0108
*DLGAP5*	DLG associated protein 5	−2.972	−6.474	0.00005	0.0001
*ASPM*	Abnormal spindle microtubule assembly	−4.055	−6.183	0.00005	0.00005
*PBK*	T-lymphokine-activated killer cell-originated protein kinase	−1.604	−6.075	0.00005	0.00025
*CCNB3*	Cyclin B3	−2.915	−6.070	0.00005	0.0002
*CCNB1*	G2/mitotic-specific cyclin-B1	−3.370	−5.934	0.00005	0.00005
*UBE2C*	Ubiquitin conjugating enzyme E2 C	−2.957	−5.859	0.00005	0.00005
*CCNB2*	G2/mitotic-specific cyclin-B2	−2.840	−5.793	0.00005	0.00005
*CDCA8*	Cell division cycle associated 8	−2.814	−5.727	0.00005	0.0001

GCs: granulosa cells; GP: gossypol; Ctrl: GP-untreated GCs; GP6.25: GCs treated with 6.25 µM GP; GP12.5: GCs treated with 12.5 µM GP.

## Data Availability

The original contributions presented in the study are included in the article/Appendix A, further inquiries can be directed to the corresponding author.

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
