# Peer review of "Effect of Gossypol on Gene Expression in Swine Granulosa Cells"

_toxins, 2024, doi:10.3390/toxins16100436_

Round 1

Reviewer 1 Report

Comments and Suggestions for Authors

Review comments onEffect of Gossypol on Gene Expression in Swine Granulosa 2 Cells”

Gossypol, a polyphenolic compound found in cotton plants, is known to have various biological effects, including hormonal and reproductive impacts. The authors chose to investigate its effect on gene expression in swine granulosa cells (which are ovarian cells involved in the maturation of eggs). Research on this topic would typically involve detailed molecular assays to identify specific genes affected by gossypol and understand the broader implications for reproductive biology. If you're exploring this area, focusing on gene expression profiles and understanding the mechanisms of how gossypol affects these pathways will be crucial for assessing its impact on swine granulosa cells.

The article is well written; a few points are there which are relevant to the study and should be considered:

The author should discuss the following in detail

  • Gossypol can interfere with steroidogenesis by affecting enzymes involved in hormone production. In granulosa cells, this might lead to altered levels of estrogen and progesterone, which are critical for reproductive function.
  • Its impact on pathways involving follicle-stimulating hormone (FSH) and luteinizing hormone (LH), essential for adequately functioning granulosa cells.
  • Gossypol could influence the expression of genes related to cell proliferation, apoptosis, and hormone production; the authors did not consider the key enzymes related to this.
  • Gossypol may affect transcription factors that regulate gene expression. This could lead to changes in the expression of genes that control cell cycle, differentiation, and response to hormonal signals; information is missing regarding this.
  • Oxidative stress caused by gossypol toxicity could affect gene expression by damaging cellular components and altering signaling pathways; the author should have mentioned that also.
  • It may induce or inhibit apoptosis (programmed cell death) in granulosa cells, influencing the expression of genes related to cell survival and death; the author should have discussed it.
  • In studies using swine granulosa cell lines, researchers might observe changes in gene expression profiles through techniques like quantitative PCR, microarrays, or RNA sequencing.
  • Fertility Impact: Changes in gene expression in granulosa cells due to gossypol exposure could have downstream effects on fertility and reproductive health in swine. This includes potential impacts on follicular development and ovulation.
  • Lack of any Functional assays to support the expression data and determine how changes in gene expression affect cellular functions such as hormone production, cell growth, and response to external stimuli.

MinorTop of Form suggestions

1.     Number lines are not systematic.

2.     Problems in scaling of figures.

3.     Why result of q-RT-PCR and RNA-seq showing a difference in the case of CYP1A1 was not justified?

4.     The bar graph for CYP1A1 in Figure 5 is not statistically significant.

5.     Density plot (Figure 2b) is not cleared. Also, the color indications for Ctrl-1 and GP12.5-2 are the same.

6.     The picture quality of Figure 4 is not good.

7.     A scatter plot of the KEGG pathway (figure 4) for control should also be added.

8.     In Figure 6 (3rd bar graph), statistics were not applied. Also, the difference is probably not statistically significant.

Comments on the Quality of English Language

Review comments onEffect of Gossypol on Gene Expression in Swine Granulosa 2 Cells”

Gossypol, a polyphenolic compound found in cotton plants, is known to have various biological effects, including hormonal and reproductive impacts. The authors chose to investigate its effect on gene expression in swine granulosa cells (which are ovarian cells involved in the maturation of eggs). Research on this topic would typically involve detailed molecular assays to identify specific genes affected by gossypol and understand the broader implications for reproductive biology. If you're exploring this area, focusing on gene expression profiles and understanding the mechanisms of how gossypol affects these pathways will be crucial for assessing its impact on swine granulosa cells.

The article is well written; a few points are there which are relevant to the study and should be considered:

The author should discuss the following in detail

  • Gossypol can interfere with steroidogenesis by affecting enzymes involved in hormone production. In granulosa cells, this might lead to altered levels of estrogen and progesterone, which are critical for reproductive function.
  • Its impact on pathways involving follicle-stimulating hormone (FSH) and luteinizing hormone (LH), essential for adequately functioning granulosa cells.
  • Gossypol could influence the expression of genes related to cell proliferation, apoptosis, and hormone production; the authors did not consider the key enzymes related to this.
  • Gossypol may affect transcription factors that regulate gene expression. This could lead to changes in the expression of genes that control cell cycle, differentiation, and response to hormonal signals; information is missing regarding this.
  • Oxidative stress caused by gossypol toxicity could affect gene expression by damaging cellular components and altering signaling pathways; the author should have mentioned that also.
  • It may induce or inhibit apoptosis (programmed cell death) in granulosa cells, influencing the expression of genes related to cell survival and death; the author should have discussed it.
  • In studies using swine granulosa cell lines, researchers might observe changes in gene expression profiles through techniques like quantitative PCR, microarrays, or RNA sequencing.
  • Fertility Impact: Changes in gene expression in granulosa cells due to gossypol exposure could have downstream effects on fertility and reproductive health in swine. This includes potential impacts on follicular development and ovulation.
  • Lack of any Functional assays to support the expression data and determine how changes in gene expression affect cellular functions such as hormone production, cell growth, and response to external stimuli.

MinorTop of Form suggestions

1.     Number lines are not systematic.

2.     Problems in scaling of figures.

3.     Why result of q-RT-PCR and RNA-seq showing a difference in the case of CYP1A1 was not justified?

4.     The bar graph for CYP1A1 in Figure 5 is not statistically significant.

5.     Density plot (Figure 2b) is not cleared. Also, the color indications for Ctrl-1 and GP12.5-2 are the same.

6.     The picture quality of Figure 4 is not good.

7.     A scatter plot of the KEGG pathway (figure 4) for control should also be added.

8.     In Figure 6 (3rd bar graph), statistics were not applied. Also, the difference is probably not statistically significant.

Reviewer 2 Report

Comments and Suggestions for Authors

Major concern 1: It is hard to draw firm conclusions due to the data from treatments with only two concentrations and one time point.

Major concern 2: When using qPCR and immunoblotting to confirm RNA-seq data, why did not include the top 10 genes with more changes listed in table 2, especially the glucose transporters?

Minor concerns:

Abstract:

Results: “Notably, genes linked to female reproductive function, cellular component organization, and oxidation-17 reduction processes exhibited differential expression in GP-treated groups.”: Need to explain more details about the major changes.

Introduction:

Line 51: Grammar error: “may be affects prepubertal reproduction”

Line 71: Need to mention the method used to evaluating cell viability

Line 83: Why did not use GP 25 as shown in Figure 1a?

Line 96: Extra space in the number “390, 951”

Line 126: Why did not choose the top 10 genes (Table 2) for qPCR confirmation?

Line 139: Why did not choose the top 10 genes (Table 2) for protein confirmation?

Methods:

Too few concentrations and time point selected for the study: “two concentrations of GP (6.25 7 and 12.5 μM) for 72 h, in vitro”

Comments on the Quality of English Language

Need some corrections on the grammars. Listed two in my comments
